# Integrating Adversarial Generative Network with Variational Autoencoders towards Cross-Modal Alignment for Zero-Shot Remote Sensing Image Scene Classification

**Suqiang Ma** [1], **Chun Liu** [1,2,3,4,*], **Zheng Li** [1,2,3] **and Wei Yang** [1,2,3]

1    The School of Computer and Information Engineering, Henan University, Kaifeng 475000, China
2    Henan Key Laboratory of Big Data Analysis and Processing, Henan University, Kaifeng 475004, China
3    Henan Engineering Laboratory of Spatial Information Processing, Henan University, Kaifeng 475004, China
4    Henan Industrial Technology Academy of Spatio-Temporal Big Data, Henan University,
     Zhengzhou 450046, China
*    Correspondence: liuchun@henu.edu.cn

**Abstract:** Remote sensing image scene classification takes image blocks as classification units and predicts their semantic descriptors. Because it is difficult to obtain enough labeled samples for all classes of remote sensing image scenes, zero-shot classification methods which can recognize image scenes that are not seen in the training stage are of great significance. By projecting the image visual features and the class semantic features into the latent space and ensuring their alignment, the variational autoencoder (VAE) generative model has been applied to address remote-sensing image scene classification under a zero-shot setting. However, the VAE model takes the element-wise square error as the reconstruction loss, which may not be suitable for measuring the reconstruction quality of the visual and semantic features. Therefore, this paper proposes to augment the VAE models with the generative adversarial network (GAN) to make use of the GAN's discriminator in order to learn a suitable reconstruction quality metric for VAE. To promote feature alignment in the latent space, we have also proposed cross-modal feature-matching loss to make sure that the visual features of one class are aligned with the semantic features of the class and not those of other classes. Based on a public dataset, our experiments have shown the effects of the proposed improvements. Moreover, taking the ResNet models of ResNet18, extracting 512-dimensional visual features, and ResNet50 and ResNet101, both extracting 2048-dimensional visual features for testing, the impact of the different visual feature extractors has also been investigated. The experimental results show that better performance is achieved by ResNet18. This indicates that more layers of the extractors and larger dimensions of the extracted features may not contribute to the image scene classification under a zero-shot setting.

**Keywords:** zero-shot learning; remote sensing image scene classification; cross-modal feature alignment; variational autoencoder; generative adversarial network

## 1. Introduction

With the rapid development of earth observation technology, there are more and more remote sensing sensors providing numerous images of the earth surface. The demand for rapid analysis of vast quantities of remote sensing images has increased significantly. Classification of these massive images is one of the most important tasks. Compared with pixel-level and object-level classification, image scene classification takes the scenes, i.e., the image blocks, as the classification units and labels them with semantic descriptors [1,2]. It is a promising way for classifying the current high-resolution remote sensing images, and plays an important role in natural disaster monitoring [3] and functional area classification [4].

Similar to image classification in the computer vision domain, the key to remote sensing image scene classification is the extraction of image scene features. Due to the

powerful feature extraction ability and successful application in various domains, deep learning has been widely applied to remote sensing image scene classification in recent years. However, these deep-learning-based scene classification methods [5–7] usually require a large number of labeled samples. Because it is difficult to obtain enough labeled samples for all classes, it is of great significance for methods which can recognize the image scenes that are not seen in the training stage. In this case, zero-shot learning (ZSL) methods which does't require any labeled samples have recently attracted much attention [8].

Derived from transfer learning [9], zero-shot learning is the training of a model using a large number of labeled samples called the seen classes of samples, and then using the model to predict the labels of unseen classes of samples, where the samples of seen and unseen classes are not identical. Taking the semantic features of the seen and unseen classes as the auxiliary information, zero-shot learning leverages the knowledge learned from the seen classes and applies them to the unseen classes. By now, many methods have emerged for zero-shot learning in the computer vision domain. The common strategy for ZSL is to build a mapping between the visual features of the seen images and the semantic features of the seen classes, and then apply the mapping to the unseen images [10,11]. This kind of methods often suffer from the hubness problem [12] or the domain shift problem [13]. Recently, more and more works are using the generative methods to tackle ZSL problem. Some works propose to utilize generative methods to generate a certain number of visual features for each unseen class conditioned on their semantic features [14]. The classes of the unseen images can be predicted by applying nearest-neighbor search algorithms. In the meantime, some works propose to apply the generative method to embed both the image visual features and the class semantic features into the latent space, and then match or align these latent features [15].

Due to the specific characteristics of remote sensing images, more challenges for zero-shot learning exist in remote sensing. For example, unlike the labels of the natural images, labels of remote sensing images can't actually reflect the semantic information of the classes. Moreover, there often exist scale variation and arbitrary orientation of geo-spatial elements in remote sensing images [16]. The images also show large intraclass differences and large interclass similarities [17,18]. With these challenges, many methods [16,19] have been proposed to address the zero-shot classification problem of remote sensing image scenes based on existing zero-shot learning works. For example, with the aim to achieve the cross-modal feature matching and address the intraclass differences and interclass similarities, a set of loss constraints has been designed in [16]; and to obtain the high-quality semantic representation of remote sensing image scenes, a remote sensing knowledge graph has been constructed [19]. In recent works, researchers have also started to use generative methods to tackle zero-shot remote sensing image scene classification. In the work of Li et al. [19], two variational autoencoders (VAE) have been utilized to project the visual features of image scenes and the semantic features of the classes into the latent space, after which the reconstruction loss and the distribution matching loss are used to achieve cross-modal feature alignment.

However, for the generative model of VAEs used for zero-shot image scene classification, an important component is the metric measuring the reconstruction quality. It calculates the element-wise square error between the true visual/semantic features and the generated visual/semantic features, to produce the reconstruction loss. This way of generating reconstruction loss may be not suitable for measuring the reconstruction quality. For example, when making some kind of transformation to the images, people may not notice the transformation; however, a large square error is produced. In light of that, this paper proposes to integrate the generative adversarial network (GAN), another kind of generative model, with the VAE models towards a cross-modal feature alignment for zero-shot remote sensing image scene classification. Our concept is augmentation of the dual VAEs with the GAN discriminator, and adopt it to learn a suitable reconstruction quality metric for VAE [20]. For two different VAE models of the dual VAEs used for two different modalities, i.e., the visual and the semantic modalities, we propose to equip each VAE

with a discriminator while using one cross-modal discriminator for both VAEs. Moreover, considering the characteristics of intraclass differences and interclass similarities among the remote sensing images, we have also designed the matching loss between the cross-modal latent features, to improve the feature alignment performance. With the cross-modal feature matching loss, our purpose is to enable the visual features of one class to be aligned with its semantic features and separated from those of other classes. We have taken the dataset presented by [19] to validate the improvements of our paper. The main contributions of this paper are as follows.

(1) We augment the VAE models with the discriminator of GAN to better measure the reconstruction quality in the zero-shot remote sensing image scene classification. Our experiments show that the discriminators contribute to the image scene classification under a zero-shot setting.

(2) We propose the cross-modal feature matching loss to address the intraclass differences and interclass similarities challenge among the remote sensing image scenes. Our experimental results have shown the effects of the cross-modal feature matching loss, and it is better to measure the matching loss with cosine distance compared with Euclidean distance and distance based on dot production.

(3) The visual features of image scenes are often extracted by different ResNet models in different works. But how about their different impact on the zero-shot image scene classification have not been investigated. Taking the three typical ResNet models of ResNet18, ResNet50, and ResNet101 for testing, our experiments show that the ResNet18 has achieved better performance. This indicates that more layers of the extractors and larger dimensions of the extracted features may not contribute to image scene classification under a zero-shot setting.

The structural arrangement of this paper is as follows: Section 2 mainly introduces the background and related works; Section 3 presents the details of the proposed method; Section 4 introduces the experiments and the results; Finally, Section 5 summarizes the conclusions of our study.

## 2. Background and Related Work

This paper involves deep learning-based generative models, zero-shot learning, and zero-shot learning for remote sensing image scene classification. In this section, we present a brief introduction about their background and related works.

### 2.1. Generative Model

The generative model mainly refers to the learning of data distribution through neural networks to generate new data. At present, there are two typical generative models, VAE and GAN.

#### 2.1.1. VAE

VAE is the variational autoencoder [21] where the autoencoder (AE) is an encoder–decoder network. In an autoencoder, the encoder maps the input data to a latent feature, and the decoder intends to reconstruct the input data from the latent feature. In this process, the distribution of the input data will be learned. Because AE has no requirement for the data distribution in the latent space and can not be used to generate new data, the variational autoencoder (VAE) is introduced to require that the data distribution in the latent space obeys the standard Gaussian distribution. Then, giving different noises, the VAE decoder can produce a variety of new data which are related to the input data.

As shown in Equation (1), the VAE loss function consists of two parts, namely, the reconstruction loss and the Kullback–Leibler (KL) divergence loss. The reconstruction loss, i.e., the first item of Equation (1), refers to the difference between the input data and the reconstructed data, where $p_E(z|x)$ indicates the distribution of the latent variable $z$ generated by the encoder, and $p_D(x|z)$ is the data distribution of reconstructed data generated by the decoder. The *KL* divergence loss, i.e., the second item of Equation (1),

measures the difference between the distribution $p_E(z|x)$ of the generated latent variable $z$ in the latent space and the Gaussian distribution unit $p(z)$.

$$loss(\theta_E, \theta_G) = E_{p_E(z|x)}[log p_D(x|z)] - D_{KL}(p_E(z|x)||p(z)) \tag{1}$$

### 2.1.2. GAN

GAN is a neural network model for generating data using an adversarial learning approach [22]. It is usually composed of a generator and a discriminator. The main task of the generator is to generate fake samples based on the random noise input. In the meantime, the discriminator's role is to judge whether the input samples are real or not. In a game-adversarial manner, the generator wants to generate real samples to cheat the discriminator, while the discriminator strives to accurately identify the fake samples generated by the generator. In the process, the generator and the discriminator constantly improve themselves until the discriminator cannot tell whether the generated sample is real or fake. It is then assumed that the generator has learned the distribution of the input data.

GAN takes the idea of minimax algorithm to construct objective function which is shown in Equation (2).

$$\min_G \max_D V(D, G) = E_{x \sim p_{data(x)}}[log D(x)] + E_{z \sim p_{noise(z)}}[log(1 - D(G(z)))] \tag{2}$$

In Equation (2), $x$ represents the real data, and $z$ represents the random noise input. $p_{data(x)}$ represents the data distribution of real data and $p_{noise(z)}$ represents the data distribution of input noise. $G(z)$ indicates the data generated by the generator when given the random noise input $z$. $D(x)$ represents the score given by the discriminator to the real data, while $D(G(z))$ is the score given to the fake data. The discriminator wants to maximize the objective function, that is, identify the fake data if at all possible. At the same time, the generator aims to minimize the objective function so that it can generate better fake data to confuse the discriminator. The parameters of the generator and the discriminator are optimized alternately and iteratively. Generally, the discriminator is optimized first before the generator.

### 2.2. Zero-Shot Learning

In daily life, humans can identify emerging objects easily based on already acquired knowledge. For example, a child has never seen a zebra, but he has seen a panda, a tiger, a horse. When he has acquired this perception that the zebra has the stripe of the tiger, the shape of the horse, and the color of the panda, he will immediately recognize the zebra when he sees the zebra. The process of reasoning about the unknown category of zebra by the known categories of tiger, horse, and panda, is that of zero-shot learning. In this process, the seen classes (i.e., panda, tiger, horse) constitute the training set, and the unseen classes (i.e., the zebra) form the test set. Meanwhile, the prior knowledge (i.e., color of pandas, stripes of tigers, shapes of horses) is the semantic information [23] related to the training set and the test set.

Zero-shot learning approaches can be divided into three categories, i.e., attribute-based zero-shot learning [24,25], embedding-based zero-shot learning [26–29], and generative model-based zero-shot learning [14,15,23,30,31]. The main idea of the attribute-based approaches is to learn an attribute classifier for each seen class and use the attribute classifiers as the space shared with unseen classes. The embedding-based methods adopt the embedding manner to map the visual features into the semantic space, semantic features into the visual space, or visual features and semantic features into a common space, and use the mappings to predict the categories of the data of unseen classes. In recent years, with the development of generative models like VAE and GAN, more and more generative methods are applied to zero-shot learning. These kinds of approaches mainly generate the

samples of unseen classes, and transform the zero-shot learning problem into a supervised learning problem.

There are two typical problems in zero-shot learning, domain shift [13] and the hubness problem [12]. Domain shift refers to the fact that the model trained from seen classes can't adapt well to unseen classes. Meanwhile, the hubness problem refers to the fact that some points will become near to most points while projecting between high dimensional spaces, resulting in poor performance when applying the nearest-neighbor searching algorithms. To address these problems, many methods have been presented in different works. For example, the SAE model [23] adds constraints to the process of embedding the semantic features into image space, to preserve the information in the image space as much as possible. Rostami et al. [32] developed a new ZSL algorithm based on coupled dictionary learning, providing attribute-aware and transductive formulations to tackle the domain-shift and the hubness challenges. Liu et al. [33] formulate a discriminative cross-aligned variational autoencoder to collect principal discriminative information from visual and semantic features to construct latent features which contain the discriminative multi-modal information associated with unseen samples.

*2.3. Zero-Shot Learning for Remote Sensing Image Scene Classification*

In recent years, zero-shot learning technology has been applied for the classification of remote sensing image scenes. For example, Chen et al. [34] proposed a zero-shot classification algorithm for remote sensing image scenes based on image feature fusion. It adopts the analytical dictionary learning method and introduces the fusion of image features to improve the zero-shot classification performance. To address the problems of the inconsistency of visual and semantic space structure and domain shift, Quan et al. [35] proposed to use the Sammon embedding and spectral clustering methods for the zero-shot remote sensing image scenes classification. Chen et al. [36] further used the analytical dictionary method to obtain the sparse coefficients of each semantic word vector, and adopted complementarity between the word vectors to obtain new word vectors.

Based on cross-domain mapping and progressive semantic benchmark modification, Li et al. [37] presented a method based on the depth feature extractor, self-coding cross-domain mapping model and modified unseen-class semantic vector to alleviate the domain drift problem. Further, Li et al. [16] proposed locality-preservation deep cross-modal embedding networks that can fully assimilate the pairwise intramodal and intermodal supervision in an end-to-end manner, so as to alleviate the problem of class structure inconsistency between two hybrid spaces. Recently, they [19] made use of the knowledge graph to enhance semantic connections between remote sensing image scene categories for the first time.

## 3. Method

In this section, we detail the proposed method of cross-modal feature alignment for zero-shot remote-sensing image scene classification. We will first give an overview of the method, and then introduce the architecture of the model integrating VAE with GAN. Finally, the training process of the proposed model is clarified.

*3.1. The Overview*

For zero-shot remote-sensing image scene classification, its purpose is to obtain a model trained from the image scenes of seen classes and use it to make correct predictions for unseen images. For this purpose, the cross-modal feature alignment method is used to train a model which projects both the visual features and semantic features of the image scenes into the latent space and make visual features in the latent space as close to their semantic features as possible. With the trained model, the visual features and semantic features of the image scenes of unseen classes can be also projected into the latent space. Then, in the latent space, a classifier will be trained with the generated semantic latent features which can classify the semantic latent features of different unseen classes. Because

the visual features in the latent space are aligned with their semantic features, the classifier obtained can be also used to classify the visual features of unseen image scenes. In this way, the classes of the unseen image scenes can be predicted. The framework of the proposed method is illustrated in Figure 1.

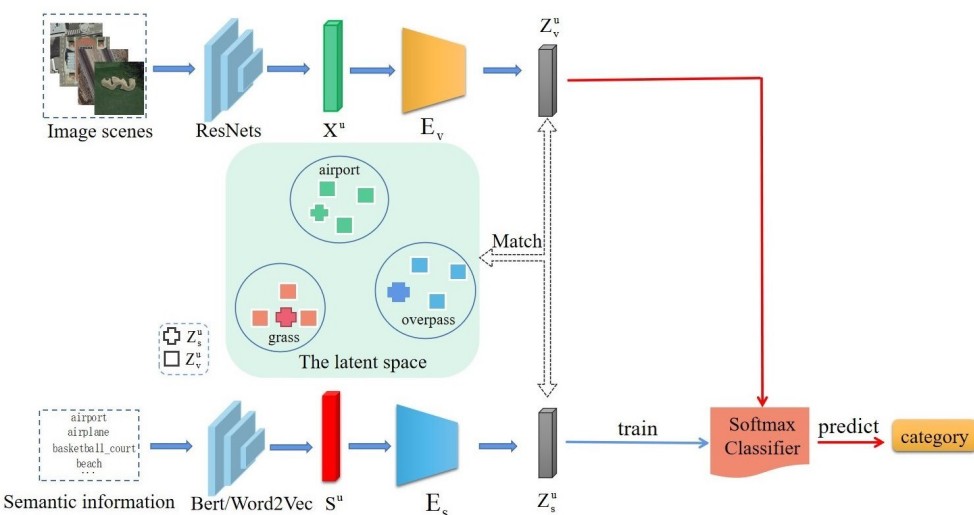

**Figure 1.** The overview of the cross-modal feature alignment method for zero-shot remote sensing image scene classification.

As shown in Figure 1, the semantic information of unseen image scene classes are embedded with the NPL model such as Word2vec [38] or Bert [39] to obtain their semantic features $s^u$. Then $s^u$ are mapped into the latent features $z_s^u$ by the encoder $E_s$. At the same time, the image scenes are embedded with a pretrained feature extractor, such as ResNet18, to obtain their visual features $v^u$, which will be further mapped into the latent features $z_v^u$ by the encoder $E_v$. In the latent space, the visual features $z_v^u$ are aligned with their semantic features $z_s^u$. That is, the visual features $z_v^u$ of each class are close to their semantic features $z_s^u$ and far from those of other classes. Using the semantic features $z_s^u$, whose classes are known, a classifier can be trained. Since the visual features in the latent space are aligned with their semantic features, such a trained classifier can be also used to classify visual features and predict their classes.

It can be seen that the key of the cross-modal feature alignment method is to obtain the two modality encoders $E_s$ and $E_v$ which project the semantic features and visual features into the latent space, respectively, and make sure that they are aligned with each other. In this paper, we propose to integrate the VAE and GAN models to train such encoders. We detail the network architecture and the training process as follows.

### 3.2. The Network Architecture

To obtain encoders which can project the visual and semantic features of image scenes into the latent space and make sure that they are aligned, this paper proposes to augment a VAE model with a GAN to integrate their strengths for better cross-modal feature alignment. The overall architecture of the integrated model can be seen in Figure 2.

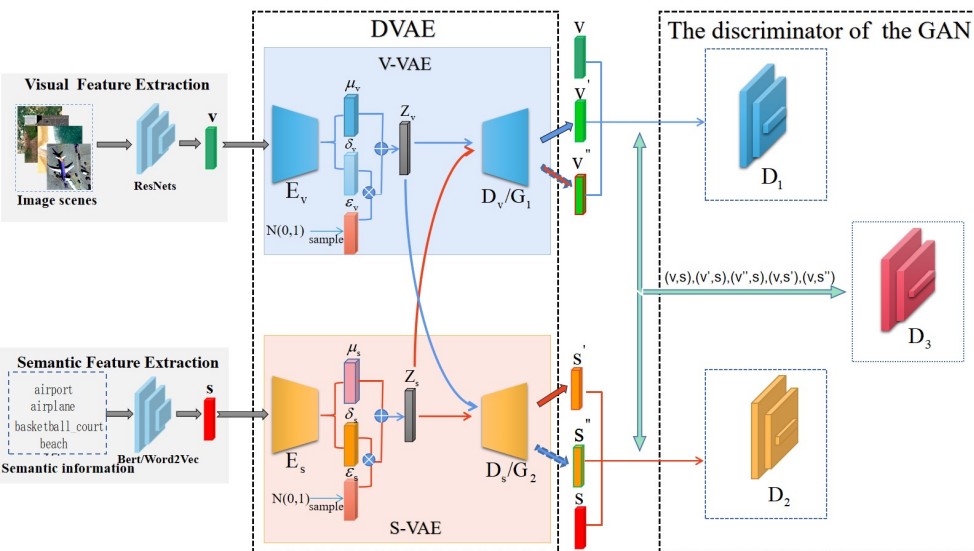

**Figure 2.** The model architecture of the proposed method.

It can be seen that there are two main components in the model, namely, DVAE and Discriminator of GAN. With two different VAEs for different modalities, the DVAE is to project the visual features and semantic features of image scenes into the latent space and then reconstruct them. Each VAE consists of an encoder and a decoder. The encoders map the visual features and the semantic features of the image scenes into the latent space. Then, the decoders reconstruct the visual features and the semantic features from the mapped latent features. The two encoders are what we want, as shown in Figure 1. In the meantime, there are three discriminators which measure the performance of the DVAE. The discriminators of $D_1$ and $D_2$ correspond to two VAEs, respectively, and the discriminator $D_3$ is for both VAEs. Particularly, the decoder $D_v$ and the discriminator $D_1$ constitute a GAN in which the decoder $D_v$ functions as the generator of the GAN. Similarly, the decoder $D_s$ and the discriminator $D_2$ constitute another GAN. In these GANs, the discriminators measure the quality of the reconstructed visual features and semantic features, and produce the probabilities that the reconstructed features are the original ones. Different from the discriminators of $D_1$ and $D_2$, the input to the discriminator $D_3$ are the cross-modal pairs of the visual features and the semantic features. It is to further judge whether the features in the pairs are the original ones or the reconstructed ones by measuring the compatibility or alignment between the cross-modal features.

Specifically, given a batch of image scenes, their visual features $v$ and corresponding semantic features $s$ are extracted in advance. Then, the encoders project them into the latent space. Following that, the mean and variance of the distributions of the visual features and the semantic features are estimated. They are the $\mu_v$, $\mu_s$, $\delta_v$, and $\delta_s$. Through sampling from the estimated distributions, the visual and semantic features $z_v$ and $z_s$ of the image scenes in the latent space are generated. For reconstruction purposes, the latent features of $z_v$ and $z_s$ will be input into the decoders $D_v$ and $D_s$ to obtain the reconstructed ones of $v'$ and $s'$. For feature alignment purposes, the latent visual features $z_v$ will be input to the semantic decoder $D_v$ to obtain the new reconstructed semantic features $s''$. At the same time, the latent semantic features $z_s$ will be input into the visual decoder $D_s$ to obtain the new reconstructed visual features $v''$. Further, all the reconstructed visual and semantic features $v'$, $v''$, $s'$, and $s''$ are input into the discriminators of $D_1$ and $D_2$ to predict the probabilities that the reconstructed features are the true features. For the discriminator $D_3$, the visual-semantic pairs of $(v,s)$, $(v,s')$, $(v,s'')$, $(v',s)$, $(v'',s)$ are input to predict the cross-modal matching degrees between the recontracted features and the true features.

Due to the application of the adversarial generative network, the proposed model will be trained step-by-step. That is, the discriminators are trained first in one epoch, and the DVAE is trained subsequently. The DVAE and the discriminators evolve through the

adversarial learning. We detail the training process of the DVAE and the discriminators as follows.

### 3.3. The Training of DVAE

Given the visual and semantic features of the image scenes, the DVAE module maps the two modality features into the latent space and then reconstructs them. With the reconstructed visual and semantic features, the discriminators measure the effect of the reconstruction. In this process, the losses are computed for training the DVAE module. In the work of Schonfeld et al. [15], three kinds of losses have been used. They are the VAE loss $L_{VAE}$, the cross-modal feature-reconstruction loss $L_{CMFR}$, and the matching loss between the visual and semantic feature distribution in the latent space $L_{VSDM}$. Besides these losses, another two kinds of losses are introduced in this paper. They are the adversarial loss $L_{ADV}$ and the cross-modal feature matching loss $L_{CMFM}$. The overall loss of the DVAE is defined as Equation (3), where $\lambda_i (i = 1, 2, 3, 4)$ is the weight factor of each kind of loss. We introduce these kinds of losses as follows.

$$L_{DVAE} = L_{VAE} + \lambda_1 L_{CMFR} + \lambda_2 L_{VSDM} + \lambda_3 L_{ADV} + \lambda_4 L_{CMFM} \tag{3}$$

#### 3.3.1. The VAE Reconstruction Loss

The VAE reconstruction loss function used in this paper is defined as Equation (4). It is the intrinsic loss function of the VAE model. By minimizing the VAE reconstruction loss, the reconstructed features are closer to the original features. In the definition, $p_E$ denotes the data distribution generated by the encoder, and the $p_D$ represents the data distribution generated by the decoder. In particular, $p_E^v(z_v|v)$ refers to the data distribution of the latent features $z_v$ generated by the encoder $E_v$ when given the visual features of $v$.

$$L_{VAE} = E_{p_E^v(z_v|v)}[log p_D^v(v|z_v)] + E_{p_E^s(z_s|s)}[log p_D^s(s|z_s)] -$$
$$D_{KL}(p_E^v(z_v|v)||p_D^v(z_v)) + D_{KL}(p_E^s(z_s|s)||p_D^s(z_s)) \tag{4}$$

#### 3.3.2. The Cross-Modal Feature Reconstruction Loss

The definition of the cross-modal feature-reconstruction loss is shown in Equation (5). Its purpose is to constrain two encoders to enable that their generated latent features to be aligned in the latent space. For this purpose, the latent semantic features are input into visual decoder to reconstruct the visual features, the latent visual features are input into semantic decoder to reconstruct the semantic features, and the distance between the reconstructed features and the original features are calculated as the loss. The $N$ denotes the number of training samples, and $v^i$ and $s^i$ represent the visual feature and semantic feature of i-th image scene.

$$L_{CMFR} = \sum_{i=1}^{N} |v^i - D_v(E_s(s^i))| + |s^i - D_s(E_v(v^i))| \tag{5}$$

#### 3.3.3. The Feature-Distribution Matching Loss

The feature-distribution matching loss is also to enable the latent feature alignment by ensuring the cross-modal distribution alignment. Its definition is shown in Equation (6), where $\mu^i$ and $\sqrt{E^i}$ represent the the mean and standard deviation of the feature distribution in the latent space corresponding to i-th image scene. Specifically, $\mu_v^i$ and $\sqrt{E_v^i}$ represent the mean and standard deviation of the distribution of visual feature.

$$L_{VSDM} = \sum_{i=1}^{N} \sqrt{||\mu_v^i - \mu_s^i||_2^2 + ||\sqrt{E_v^i} - \sqrt{E_s^i}||_2^2} \tag{6}$$

### 3.3.4. The Adversarial Loss

The adversarial loss comes from the discriminators. When inputting the reconstructed features from the DVAE into the discriminators of $D_1$ and $D_2$, we expect that the discriminators can't recognize them as the reconstructed ones from the perspective of the decoders of $D_v$ and $D_s$. This means that we expect them to predict the probabilities that the reconstructed features are the original ones as much as possible. Thus, the adversarial losses from the discriminators of $D_1$ and $D_2$ are defined as follows.

$$L_{ADV}(D_1, D_2) = E[D_1(v') - 1]^2 + E[D_1(v'') - 1]^2 + E[D_2(s') - 1]^2 + E[D_2(s'') - 1]^2 \tag{7}$$

For the discriminator $D_3$, the inputs are the visual-semantic pairs of $(v, s')$, $(v, s'')$, $(v', s)$, and $(v'', s)$. We also expect that the discriminator will not recognize these input features as the reconstructed ones and predict the probabilities that the features are compatible as much as possible. Thus, the adversarial loss function for the discriminator of $D_3$ is defined as follows.

$$L_{ADV}(D_3) = E[D_3(v, s') - 1]^2 + E[D_3(v, s'') - 1]^2 + \\ E[D_3(v', s) - 1]^2 + E[D_3(v'', s) - 1]^2 \tag{8}$$

Therefore, the total adversarial losses coming from the discriminators are as follows:

$$L_{ADV} = L_{ADV}(D_1, D_2) + L_{ADV}(D_3) \\ = E[D_1(v') - 1]^2 + E[D_1(v'') - 1]^2 + E[D_2(s') - 1]^2 + E[D_2(s'') - 1]^2 + \\ E[D_3(v, s') - 1]^2 + E[D_3(v, s'') - 1]^2 + E[D_3(v', s) - 1]^2 + E[D_3(v'', s) - 1]^2 \tag{9}$$

### 3.3.5. The Cross-Modal Feature Matching Loss

Considering the characteristics of intraclass differences and interclass similarities among the remote sensing image scenes, we introduce the cross-modal feature matching loss to further narrow the intraclass differences between the visual features and the semantic features of the same classes in the latent space, and enlarge the interclass distance between the visual features and the semantic features of different classes. The definition of the cross-modal feature matching loss is shown in Equation (8) where $c_{z_v^i}$ denotes the class of the i-th latent feature of $z_v^i$, and *cos* means the matching metric of cosine distance. $N_1$ is the number of the cross-modal feature pairs in which both the visual feature and the semantic feature come from the same classes. Meanwhile, $N_2$ is the number of the cross-modal feature pairs in which the visual feature and the semantic feature are from different classes.

$$L_{CMFM} = \frac{1}{N_1} \sum_{c_{z_v^i} = c_{z_s^j}} cos(z_v^i, z_s^j) - \frac{1}{N_2} \sum_{c_{z_v^i} \neq c_{z_s^j}} cos(z_v^i, z_s^j) \tag{10}$$

### *3.4. The Training of Discriminators*

Once the visual features and the semantic features are reconstructed by the decoders of the VAEs, the discriminators measure the performance of the reconstructed features by judging whether there are reconstructed features in the inputs. Particularly, when training the discriminator of $D_1$, we input the original visual features $v$ and the reconstructed visual features $v'$ and $v''$ into the discriminator of $D_1$. And we expect that the discriminator has the ability to distinguish the reconstructed visual features $v'$ and $v''$ from the original visual features $v$. That is, we expect that the discriminator produces the probability that the input features are the original ones as much as possible for $v$, but, on the contrary, as small as possible for $v'$ and $v''$. Thus, the loss function for the discriminator of $D_1$ can be defined as follows.

$$L_{ADV}(D_1) = E[D_1(v) - 1]^2 + E[D_1(v')]^2 + E[D_1(v'')]^2 \tag{11}$$

Similarly, the loss function of the discriminator of $D_2$ can be defined as follows.

$$L_{ADV}(D_2) = E[D_2(s) - 1]^2 + E[D_2(s')]^2 + E[D_2(s'')]^2 \tag{12}$$

For the discriminator of $D_3$, we will input the pairs of $(v, s)$, $(v, s')$, $(v, s'')$, $(v', s)$, and $(v'', s)$ into it for training. The discriminator of $D_3$ will produce the probabilities indicating whether there are reconstructed features in the pairs. Then, when inputting the pairs of $(v, s)$, we expect that the discriminator of $D_3$ predicts the portability as much as possible. In contrast, we expect that the discriminator of $D_3$ predicts the probabilities as small as possible for these kinds of pairs of $(v, s')$, $(v, s'')$, $(v', s)$, and $(v'', s)$. Therefore, the loss function of the discriminator of $D_3$ can be defined as follows:

$$L_{ADV}(D_3) = E[D_3(v, s) - 1]^2 + E[D_3(v, s')]^2 + E[D_3(v, s'')]^2 \\ + E[D_3(v', s)]^2 + E[D_3(v'', s)]^2 \tag{13}$$

Finally, the loss function for training these three discriminators is as follows:

$$L_{ADV}(D_1, D_2, D_3) = L_{ADV}(D_1) + L_{ADV}(D_2) + L_{ADV}(D_3) \tag{14}$$

## 4. Experiments

In this section, extensive experiments are conducted to evaluate the effectiveness of the proposed method by attempting to answer the following four research questions.

RQ1. How do the different kinds of losses defined in Equation (3) contribute to model performance?

RQ2. Does the proposed method achieve better performance when compared with related methods?

RQ3. Does each improvement, i.e., the three discriminators and the cross-modal feature-matching loss, actually work as expected?

RQ4. What is the impact of different visual feature extractors on zero-shot image scene classifications performance?

### 4.1. Experimental Setup

4.1.1. Data for Experiments

This paper takes the dataset which has been used in the work of Li, et al. [19] for experiments. This dataset is the integration of five public remote sensing image scenes datasets including UCM [40], AID [41], NWPU-RESISC [42], RSI-CB256 [43], and PatternNet [44]. The merged dataset realizes the complementarity between different classes and increases the diversity. This contributes to the validation of the zero-shot classification performance. There are 70 classes in the dataset, and 800 images with the size of 256 pixel $\times$ 256 pixel for each class. Some images of the dataset are shown in Figure 3.

4.1.2. Metric of the Experiment

To measure the performance of the proposed method, this paper adopts the metric of overall accuracy (OA) which is defined as follows.

$$OA = \frac{1}{m} \sum_c^m \frac{correct\ predictons\ in\ c}{total\ samples\ in\ c} \tag{15}$$

In the above equation, we adopt the widely used average per-class top-1 accuracy to evaluate the performance of each model where $m$ represents the number of unseen classes. For each class of image scenes for testing, the accuracy is calculated by dividing the number of image scenes correctly classified by its total number of image scenes. The overall accuracy is the average of the accuracy of each class.

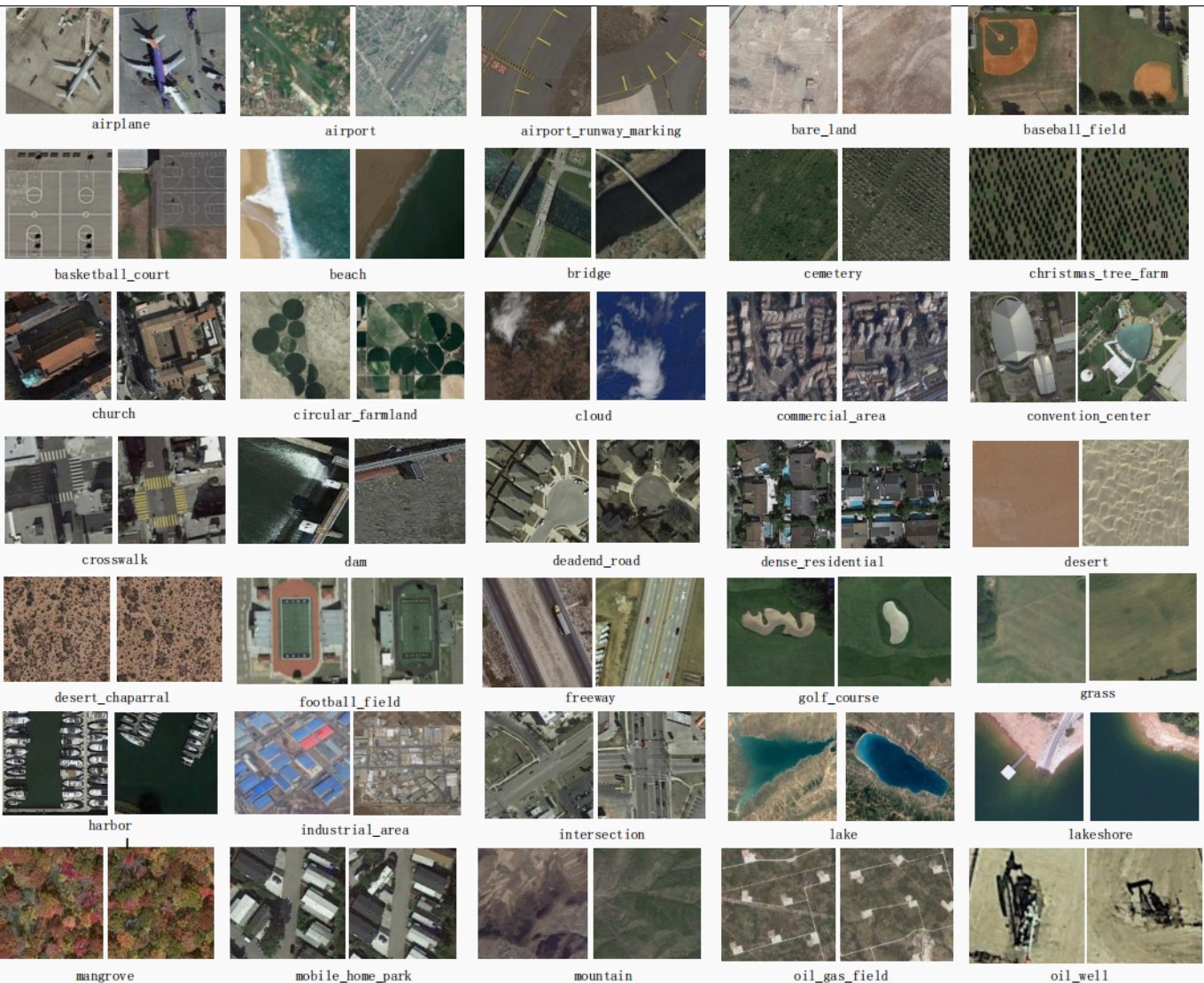

**Figure 3.** The examples of the remote sensing image scenes of the dataset.

### 4.1.3. The Implementation

For the implementation of the proposed method, all the encoders and decoders are neural networks with only one layer. For the encoder and the decoder of the visual modality, their dimensions are set to 512. As the semantic feature dimension is smaller than that of visual features, the dimensions of the encoder and decoder for the semantic modality are set to 256. The discriminator $D_1$ is designed as a neural network with only one hidden layer where the dimension is 1200. The discriminator $D_2$ is also designed as a neural network with only one hidden layer where the dimension is 256. The discriminator $D_3$ is designed as a neural network with two fully connected layers, where the dimensions are 1200 and 600. We set the batchsize to 50, the dimension of the latent feature vector to 32, and take 50 epochs to train the model. When using the generated latent semantic features to train a classifier for predicting the classes of unseen image scenes, the softmax classifier is applied. Our implementation is based on that of [15].

We used the classical CNN backbone of the ResNet network, such as ResNet101, ResNet50, and ResNet18 models [45], to extract the visual features of image scenes. They are pretrained on the ImageNet dataset. The 2048-dimensional features are extracted from the remote-sensing image scenes by using ResNet50 and ResNet101, and 512-dimensional features are there when using ResNet18. Regarding the extraction of semantic features, we adopted two kinds of sematic features used by the work of Li et al. [19] for experi-

ments, (1) the 300-dimensional features which are extracted from the class labels by using Word2Vec; (2) the 1024-dimensional features which are extracted by using BERT from a set of sentences describing the image scene classes.

### 4.2. Hyperparameter Analysis

To evaluate the effect of the hyperparameters and answer RQ1, we analyzed the sensitivity of the hyperparameters of $\lambda_1$, $\lambda_2$, $\lambda_3$, and $\lambda_4$ shown in Equation (3). They are the weight factors of cross-modal feature reconstruction loss, feature-distribution-matching loss, adversarial loss, and cross-modal feature-matching loss. The experiment has been conducted under the seen/unseen ratio of 60/10, and the visual features from ResNet18 and the semantic features from Bert are adopted. The average of the classification accuracies over five random seen/unseen splits are recorded.

As shown in Figure 4, while setting $\lambda_2$, $\lambda_3$, and $\lambda_4$ to 1, we tested the values in the set of $[0.001, 0.01, 0.1, 1, 10, 100]$ for $\lambda_1$, and found that best performance was achieved when $\lambda_1 = 1$. In the same way, we set $\lambda_1$, $\lambda_3$, and $\lambda_4$ as 1, our proposed method achieved the best performance when $\lambda_2 = 0.01$. By setting $\lambda_1$ as 1, $\lambda_2$ as 0.01, and $\lambda_4$ as 1 to test the values of $\lambda_3$, best performance was there when $\lambda_3 = 0.1$. When testing the values of $\lambda_4$, we set $\lambda_1$ as 1, $\lambda_2$ as 0.01, and $\lambda_3$ as 0.1, with the results indicating that it is better to set $\lambda_4$ to 0.1. Therefore, in our following experiments, we have set the hyperparameters of $\lambda_1$, $\lambda_2$, $\lambda_3$, and $\lambda_4$ shown in Equation (3) to 1, 0.01, 0.1, and 0.1, respectively.

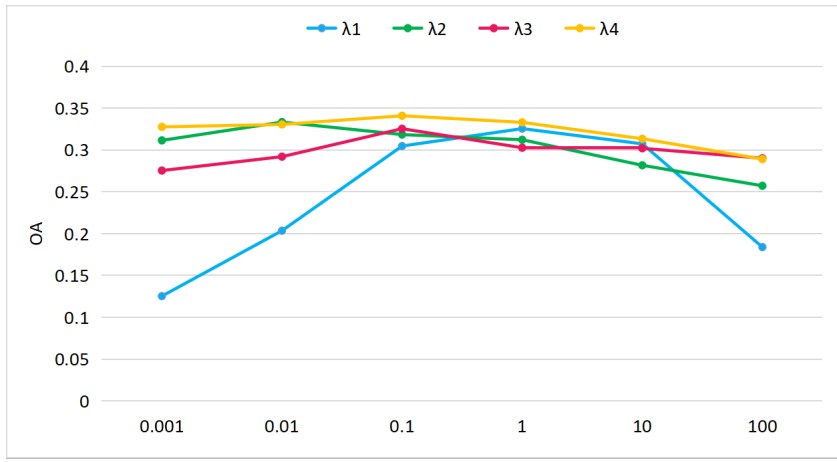

**Figure 4.** The sensitivity analysis of the weight factor assigned to the different losses.

### 4.3. Comparison with Related Methods

To validate the performance of our method and answer RQ2, we have compared it with several classical zero-shot methods. These methods include the embedding-based method of SPLE [46] and the generative model-based methods of SAE [23], CIZSL [47], GDAN [14], and CADA-VAE [15]. SPLE [46] introduced the idea of semantically preserving positional embedding, and achieved better matching between visual features and semantic features. SAE [23] adopted the semantic feature representation as the hidden layer, and followed the AE model to learn the mapping from semantic space to visual space. CIZSL [47] used generative adversarial network for zero-shot learning and introduced hallucinated text to the generator, encouraging the generated visual features to deviate from the seen classes and thus making the generated samples more diverse. GDAN [14] built a dual generative adversarial networks and used dual adversarial loss and cycle consistency loss to bidirectionally map visual and semantic features. CADA-VAE [15] proposed the construction of visual and semantic variational autoencoders to reconstruct features and align them in the latent space so that the constructed features contain basic multimodal information related to unseen classes. We have divided the dataset according to the ratios of 60/10, 50/20, and 40/30 to obtain the training set and the testing set. And both the

semantic features from Word2vec and Bert are taken into consideration. The results of the comparison are shown in Table 1.

**Table 1.** The results of the comparison with related zero-shot methods.

| Semantic Features Types | Word2Vec | | | Bert | | |
|---|---|---|---|---|---|---|
| Seen/Unseen Ratios | 40/30 | 50/20 | 60/10 | 40/30 | 50/20 | 60/10 |
| SPLE [46] | 9.8% ± 1.4% | 13.2% ± 1.9% | 20.1% ± 3.7% | 8.3% ± 2.0% | 13.2% ± 2.6% | 19.0% ± 3.8% |
| SAE [23] | 9.6% ± 1.4% | 13.7% ± 1.7% | 23.5% ± 4.2% | 8.8% ± 1.3% | 12.4% ± 1.9% | 22.0% ± 1.7% |
| CIZSL [47] | 6.0% ± 1.2% | 10.6% ± 3.7% | 20.6% ± 0.4% | 6.2% ± 2.1% | 10.3% ± 1.9% | 20.4% ± 4.1% |
| GDAN [14] | 8.1% ± 1.1% | 13.6% ± 4.7% | 27.6% ± 5.9% | 9.8% ± 0.9% | 13.8% ± 1.1% | 32.1% ± 8.2% |
| CADA-VAE [15] | 8.23% ± 2.7% | 14.68% ± 1.87% | 23.37% ± 1.85% | **11.45% ± 0.36%** | 15.56% ± 1.3% | 32.85% ± 2.51% |
| **Ours** | **9.31% ± 1.56%** | **16.56% ± 3.5%** | **27.98% ± 1.35%** | 11.42% ± 0.42% | **17.86% ± 1.09%** | **35.08% ± 1.23%** |

As can be seen from Table 1, our method achieved the optimal performance in most cases, which showed its effectiveness. Generally speaking, the embedding-based method of SPLE has not shown competitive performance when compared with these generative model-based methods. But, among these generative-based methods, the CIZSL method has the worst results. This may be because that the remote sensing image scenes are complex, usually containing a variety of objects. The generative adversarial network may not generate high-quality samples well due to its training instability, resulting in unsatisfactory ZSL classification results. Meanwhile, compared with the methods of SAE and GDAN, CADA-VAE has achieved better performance. This may be due to the fact that, unlike SAE and GDAN, CADA-VAE adopts the cross-modal latent feature alignment for zero-shot image scene classification instead of following the generative models to generate samples for unseen classes. Our method improved the CADA-VAE method by augmenting the VAEs with the discriminators of GAN, and the cross-modal feature matching loss. It can be seen that our method outperforms the CADA-VAE method, which validates the contribution of our improvements.

Moreover, when comparing the results under the semantic features from Word2vec and Bert, our method has better performance when adopting the semantic features from Bert, especially under the dividing ration of 60/10. This may be because that the semantic features from Bert with 1024 dimensions contain more information about the characteristics of image scenes. Further, similar to other methods, the proposed method obtained the best results under the dividing ratio of 60/10. This is because there are more classes for training and more knowledge can be leveraged for unseen classes.

### 4.4. Ablation Experiments

There are several new components (i.e., the discriminators of $D_1$, $D_2$, $D_3$, and the cross-modal matching loss) we have introduced, compared with the CADA-VAE method [15]. To validate the benefits of these components so as to answer RQ3, we conducted ablation experiments by using semantic features from Bert. In the experiment, we have also validated the ways of dot production and Euclidean distance for calculating the cross-modal matching loss as well as the cosine distance. As shown in Table 2, we have constructed the following model variants.

DVAE-DGAN augment the dual VAEs model with two discriminators of $D_1$ and $D_2$, that is, equipping each VAE with one discriminator.

DVAE-GAN augment the dual VAEs model with one single cross-modal discriminator of $D_3$ for both VAEs.

DVAE-TGAN augment the dual VAEs model with all the discriminators of $D_1$, $D_2$, and $D_3$.

DVAE-TGAN-$L_{CMFM_1}$, while adopting all the discriminators $D_1$, $D_2$, and $D_3$, adopt the dot production to calculate the cross-modal matching loss.

DVAE-TGAN-$L_{CMFM_2}$ while adopting the discriminators $D_1$, $D_2$, and $D_3$, adopt the Euclidean distance to calculate the cross-modal matching loss.

DVAE-TGAN-$L_{CMFM_3}$ while adopting the discriminators $D_1$, $D_2$, and $D_3$, adopt the cosine distance to calculate the cross-modal matching loss.

**Table 2.** Ablation experiments.

| Variants | $D_1 + D_2$ | $D_3$ | Dot_production | Euclidean_distance | Cosine_distance | 40/30 | 50/20 | 60/10 |
|---|---|---|---|---|---|---|---|---|
| DVAE-DGAN | ✓ | ✗ | ✗ | ✗ | ✗ | $10.67\% \pm 0.83\%$ | $15.74\% \pm 0.01\%$ | $34.16\% \pm 2.57\%$ |
| DVAE-GAN | ✗ | ✓ | ✗ | ✗ | ✗ | $10.22\% \pm 0.85\%$ | $15.49\% \pm 0.6\%$ | $32.3\% \pm 2.41\%$ |
| DVAE-TGAN | ✓ | ✓ | ✗ | ✗ | ✗ | $11.3\% \pm 0.79\%$ | $16.2\% \pm 1\%$ | $34.42\% \pm 1.75\%$ |
| DVAE-TGAN-$L_{CMFM_1}$ | ✓ | ✓ | ✓ | ✗ | ✗ | $9.79\% \pm 0.47\%$ | $16.92\% \pm 1.46\%$ | $34.48\% \pm 0.54\%$ |
| DVAE-TGAN-$L_{CMFM_2}$ | ✓ | ✓ | ✗ | ✓ | ✗ | $10.89\% \pm 0.63\%$ | $16.99\% \pm 1.77\%$ | $34.79\% \pm 1.2\%$ |
| DVAE-TGAN-$L_{CMFM_3}$ | ✓ | ✓ | ✗ | ✗ | ✓ | $\mathbf{11.42\% \pm 0.42\%}$ | $\mathbf{17.86\% \pm 1.09\%}$ | $\mathbf{35.08\% \pm 1.23\%}$ |

It can be seen from Table 2 that DVAE-TGAN achieves best result compared with the variants of DVAE-DGAN and DVAE-GAN. This shows the benefits of all these discriminators. When applying all these discriminators, more constraint information will be obtained. The constraint information can improve the encoders to better map the visual and semantic features of image scenes, and the decoders to better reconstruct them. Meanwhile, it is obvious that the strategy of equipping each VAE with a discriminator (i.e., DVAE-DGAN) is superior to that of applying a single cross-modal discriminator for both VAEs (i.e., DVAE-GAN). The reason may be that the different discriminators for different modalities will measure the reconstruction error more accurately than the cross-modal discriminator.

In addition, when comparing DVAE-TGAN with the variants of DVAE-TGAN-$L_{CMFM_1}$, DVAE-TGAN-$L_{CMFM_2}$, and DVAE-TGAN-$L_{CMFM_3}$, it can be seen that better performance is provided by the cross-modal feature-matching loss in most cases. This demonstrates the effectiveness of the cross-modal feature-matching loss. Among these methods of calculating the cross-modal feature-matching loss, it is better to use the cosine distance. This may be because it is better able to validate the alignment between the cross-modal features which are located in the high-dimensional space.

### 4.5. The Impact Evaluation of the Visual Feature Extractors

To achieve the cross-modal feature alignment in the latent space, the visual and semantic features are often extracted in advance by applying some extractors. It is obvious that these extractors play an important role for zero-shot image scene classification. The impact of the semantic feature extractors such as Word2Vec and Bert has been evaluated in related works. However, the impact of these visual feature extractors has not been investigated. Thus, taking the extractors of ResNet18, ResNet101 and ResNet50 as example, we have done an experiment to validate their different impact on our method and answer the research question RQ4. Figure 5 shows the results of this experiment. It can be seen that for both two kinds of semantic features, better performance is there when ResNet18 is used as the visual feature extractor. In comparison, the features extracted by ResNet18 are smaller in dimension, i.e., 2048 dimensions for ResNet50 and ResNet101, and 512 dimensions for ResNet18. And from Resnet18 to Resnet50 and to Resnet101, there are more and more layers in each model. The results shown in Figure 5 indicate that larger dimensions and more layers may not contribute to the image scene classification under a zero-shot setting. This may be because the shallow features will make a greater contribution to zero-shot image scene classification but are lost with the increase of layers.

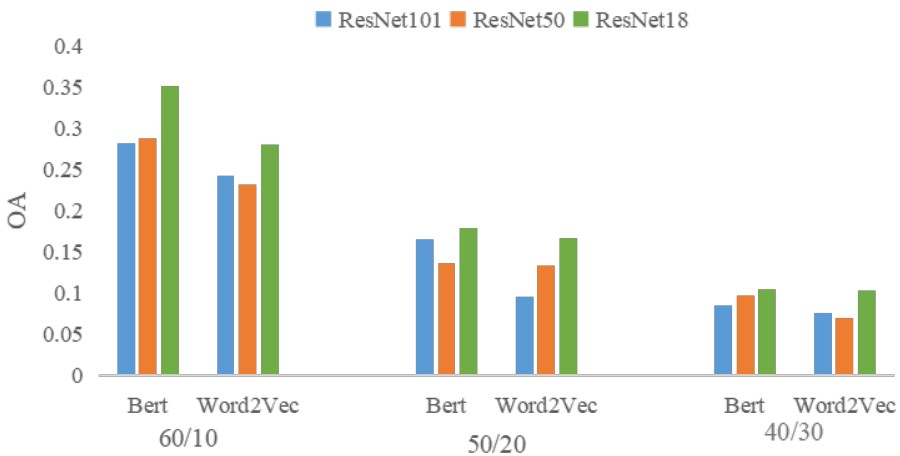

**Figure 5.** The results of the experiment of evaluating the impact of different visual feature extractors.

*4.6. Visualization of the Latent Visual and Semantic Feature Alignment*

The proposed method attempts to achieve better feature alignment between the visual and semantic modalities, that is, each class of visual features are closer to their own semantic feature and farther from other classes of semantic features. In order to provide a qualitative evaluation of the proposed method, we have visualized the latent visual and semantic features of the unseen image scenes to observe the cross-modal feature alignment in the latent space. Figure 6 shows one of the results, where Figure 6a is the visualization of cross-modal feature alignment without our improvements, while Figure 6b is the visualization of the results of our method. Since there are more unseen classes under the seen/unseen ratios of 50/20 and 40/30, we have only used the latent features under the seen/unseen ratio of 60/10 for better visualization. Thus, there are 10 unseen classes of image scenes in the visualization. And there are 800 latent visual features (denoted by **.**) and 1 latent semantic feature (denoted by ▲) for each unseen class. The original visual features of the image scenes are extracted by ResNet18 and the original semantic features are extracted by Bert. The tool of t-SNE [48] is used for the cross-modal feature alignment visualization.

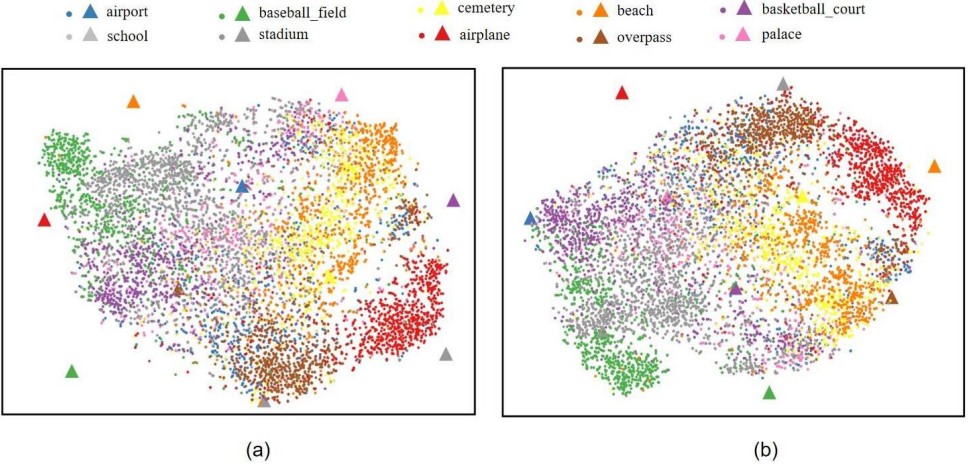

**Figure 6.** T-SNE visualization of the cross-model feature alignment between the visual (**.**) and semantic (▲) features of unseen image scenes in the latent space.

From Figure 6a, we can see that the latent visual features of the classes of basketball _court (i.e., purple color), palace (i.e., pink color), and school (i.e., light grey color) are far apart from the latent semantic features of these classes. By comparison, as shown in Figure 6b, the latent semantic features of these classes are surrounded by the latent visual features. In addition, it can be also seen that for the classes of baseball_field (i.e., green color),

airplane (i.e., red color), and beach (i.e., orange color), the latent visual features become closer to the latent semantic features of their classes after applying the proposed method. But for the classes of airport (i.e., blue color) and overpass (i.e., khaki color), it seems that the latent visual features of these classes become further from the semantic features of these classes. On the whole, these visualization results also indicate the contribution of our proposed method to the cross-modal feature alignment.

## 5. Conclusions

This paper proposes augmentation of the dual VAEs with a GAN, for cross-modal feature alignment for zero-shot remote-sensing image scene classification. The concept is to make use of the GAN's discriminator in order to learn a suitable reconstruction quality metric for the VAE. Given that there are two VAEs for the visual and semantic modalities, respectively, we propose to equip each VAE with a discriminator while adding another cross-modal discriminator for both VAEs. To promote feature alignment in the latent space and address the challenge of intraclass differences and interclass similarities, we have also proposed the cross-modal feature-matching loss to make sure that the visual features of one class are aligned with the semantic features of the class and unaligned with those of other classes. Based on the public dataset, our experiments have shown the contributions of the discriminators and the cross-modal feature-matching loss. In light of the fact that the visual and semantic features are often extracted in advance by applying some extractors and the impact of the different visual feature extractors has not been investigated, we have taken the ResNet models of ResNet18, extracting 512-dimensional visual features, and ResNet50 and ResNet101, both extracting 2048-dimensional visual features, for testing. The experimental results show that better performance is achieved by ResNet18, which indicates that more layers of the extractors and larger dimensions of the extracted features may not contribute to the image scene classification under zero-shot setting.

**Author Contributions:** Conceptualization, S.M. and C.L.; methodology, S.M.; software, S.M.; validation, S.M., C.L. and Z.L.; formal analysis, S.M.; investigation, S.M.; resources, W.Y.; data curation, Z.L.; writing—original draft preparation, S.M.; writing—review and editing, S.M., C.L. and W.Y.; visualization, S.M. and W.Y.; supervision, C.L., W.Y. and Z.L.; project administration, C.L. and Z.L.; funding acquisition, C.L. All authors have read and agreed to the published version of the manuscript.

**Funding:** This research received no external funding.

**Data Availability Statement:** Not applicable.

**Conflicts of Interest:** The authors declare no conflict of interest.

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
