# Peer review of "Integrating Adversarial Generative Network with Variational Autoencoders towards Cross-Modal Alignment for Zero-Shot Remote Sensing Image Scene Classification"

_remotesensing, doi:10.3390/rs14184533_

Round 1
Reviewer 1 Report (Previous Reviewer 3)
This is a re-submitted version and the authors have well addressed my comments.
Reviewer 2 Report (Previous Reviewer 2)
The authors have addressed my concerns by improving the experimental results and revising the presentation.
Reviewer 3 Report (Previous Reviewer 1)
It´s nice work.
This manuscript is a resubmission of an earlier submission. The following is a list of the peer review reports and author responses from that submission.
Round 1
Reviewer 1 Report
Minor errors in the manuscript indicated with different colors.

Author Response
Dear Reviewer,
Thank you very much for your time involved in reviewing the manuscript and your very encouraging comments.
We also appreciate your clear and detailed feedback and based on your suggestions, we have carefully revised our manuscript, which is of great help to our article. Please see the PDF of the manuscript for all revisions.
We would like to take this opportunity to thank you for all your time involved and this great opportunity for us to improve the manuscript. We hope you will find this revised version satisfactory.
Sincerely,
The Authors

Reviewer 2 Report
In this paper, an algorithm for zero-shot learning of classes based on using semantic descriptions in the remote sensing domain has been developed. The idea is to use generative adversarial networks to align the semantic descriptions with visual features in the bottleneck of a variational autoencoder and improve the reconstruction loss for the variational autoencoder. Experiments using three network architectures on a real-world dataset are provided to demonstrate that the proposed method is effective and the code is released for public use.
The paper uses state-of-the-art ideas from artificial intelligence to address the critical challenge of zero-shot learning in remote sensing domains. The results are also convincing. I have the following comments to be addressed before publication. 1. Although there are extensive experiments with varying coefficients in Eq 3, an ablative experiment on Eq 3 can be informative. Please performs a set of experiments by letting \lambda_i =0 in Eq 3 for all coefficients to study the effect of each loss term and its importance on the eventual performance. 2. Comparison against prior works is missing which makes it difficult to judge how well the proposed method performs. I understand that the proposed method works on a new domain but I think it is necessary to use 2-3 previous algorithms on the dataset to demonstrate the competitiveness of the proposed algorithms. 3. In Figures 4 and 5, please visualize the standard deviations for improved comparison. The numbers are close and this will help to see if the differences are statistically significant. 4. The problems of domain shift and hubness in zero-shot learning has been studied before. For example. a. Dinu, G., Lazaridou, A. and Baroni, M., 2014. Improving zero-shot learning by mitigating the hubness problem. arXiv preprint arXiv:1412.6568. b. Kodirov, E., Xiang, T. and Gong, S., 2017. Semantic autoencoder for zero-shot learning. In Proceedings of the IEEE conference on computer vision and pattern recognition (pp. 3174-3183). c. Liu, Y., Gao, X., Han, J. and Shao, L., 2022. A Discriminative Cross-Aligned Variational Autoencoder for Zero-Shot Learning. IEEE Transactions on Cybernetics. d. Rostami, M., Kolouri, S., Murez, Z., Owechko, Y., Eaton, E. and Kim, K., 2022. Zero-shot image classification using coupled dictionary embedding. Machine Learning with Applications, 8, p.100278. I think it is helpful to include the above work in the Background and Related Work section to give the reader a broader perspective about prior works that address these two important challenges. 5. Could you expand Figure 7 by adding a subfigure to represent data when the GAN network is not used? I would like to see how is the effect of using GAN to make the embeddings more discriminative.Author Response
Please see the attachment.

Reviewer 3 Report
please see attached file for more details.

Round 2
Reviewer 2 Report
The authors have addressed some of my concerns but I still think more efforts can be attempted.
Point 1: I don't think being time-consuming is a good reason to differ the experiments to the future. Please spend time and do these experiments, even in a simplified format.
Point 2: Again, just pushing something to the future is not a convincing reason.
Point 4: Please include all four references in my original comment and be more comprehensive to explain the differences and do some comparison as a paragraph.
Please address the above and resubmit again.
Reviewer 3 Report
I have no further comments.